# Dengue-Associated Hemophagocytic Lymphohistiocytosis: A Narrative Review of Its Identification and Treatment

**DOI:** 10.3390/pathogens13040332

**Published:** 2024-04-17

**Authors:** Kay Choong See

**Affiliations:** Division of Respiratory and Critical Care Medicine, Department of Medicine, National University Hospital, Singapore 119228, Singapore; kaychoongsee@nus.edu.sg

**Keywords:** critical care, cytokine release syndrome, flavivirus, intensive care units, multiple organ failure, severe dengue

## Abstract

Dengue’s lack of specific treatments beyond supportive care prompts a focus on uncovering additional pathophysiological factors. Dengue-associated hemophagocytic lymphohistiocytosis (HLH), characterized by dysregulated macrophage activation and cytokine storm, remains underexplored despite its potential to worsen disease severity and mortality. While rare, dengue-associated HLH disproportionately affects severe cases, significantly impacting mortality rates. To mitigate high mortality, early identification and familiarity with dengue-associated HLH are imperative for prompt treatment by clinicians. This narrative review therefore aims to examine the current clinical and therapeutic knowledge on dengue-associated HLH, and act as a resource for clinicians to improve their management of HLH associated with severe dengue. Dengue-associated HLH should be considered for all cases of severe dengue and may be suspected based on the presence of prolonged or recurrent fever for >7 days, or anemia without intravascular hemolysis or massive bleeding. Diagnosis relies on fulfilling at least five of the eight HLH-2004 criteria. Treatment predominantly involves short courses (3–4 days) of high-dose steroids (e.g., dexamethasone 10 mg/m^2^), with additional therapies considered in more severe presentations. Notably, outcomes can be favorable with steroid therapy alone.

## 1. Introduction

Dengue is one of the most important vector-borne diseases globally, affecting large geographical regions and contributing to symptomatic disease in 60–100 million new cases annually [1,2]. Each year, the estimated global impact of dengue includes 10,000 deaths, half a million years of life lost to premature mortality, half of million years lived in disability, a million disability-adjusted life-years, and costs of USD 8.9 billion [1,2,3].

Mosquito control, the use of personal protection, and vaccination are means for preventing dengue infection. Three tetravalent vaccines have demonstrated efficacy against dengue infection in Phase III clinical trials: the three-dose CYD-TDV (Dengvaxia^®^, Sanofi S.A. Paris, France) [4,5], the two-dose TAK-003 (Qdenga^®^, Takeda Pharmaceutical Company Limited, Tokyo, Japan) [6,7], and the one-dose Butantan-DV (Instituto Butantan, São Paulo, Brazil) [8]. Of note, CYD-TDV paradoxically increased the risk of severe disease in seronegative recipients through unknown immunopathologic mechanisms, and thus should only be given to persons with at least one previous dengue infection [9].

Once an individual is infected, the overall management of dengue is supportive [10]. Capillary leak and hypotension are corrected with fluid therapy, bleeding with blood product support, cardiomyopathy with inotropic agents to increase cardiac function, and respiratory failure with mechanical ventilation. Although good supportive care may mitigate the poor outcomes of dengue, the lack of any specific treatment limits the potential for further improvement of health outcomes. Efforts directed at uncovering and treating the underlying pathophysiology of severe dengue are therefore urgently needed.

As severe dengue is marked by increased plasma leakage, much research has aimed to preserve capillary permeability [11]. Nonetheless, another aspect of severe dengue—that of dengue-associated hemophagocytic lymphohistiocytosis (HLH)—may be overlooked, and it is not mentioned in the World Health Organization dengue management guidelines in 2009 [10].

HLH, in general, is a rare diagnosis that may be associated with a variety of infectious triggers (especially Epstein–Barr virus [12], dengue virus [13], and scrub typhus [14]), autoimmune disorders [15,16,17], malignancy [18,19,20], and familial genetic mutations [21]. It is predominantly found in children, with an incidence of about 1 in 100,000 hospitalized children [22]. Adult cases are increasingly recognized [23,24]. As a subset of HLH, dengue-associated HLH is even more uncommon among all dengue patients, occurring with an incidence of 0.44 cases per 100,000 children per year [25]. Nonetheless, dengue-associated HLH is found more commonly in severe dengue patients; it was observed in 15 (13.4%) of 112 children with severe dengue [26] and in 5 (38.5%) of 13 non-survivors in a study of 114 patients admitted to an adult intensive care unit [27]. Unlike severe dengue, which has no known specific treatment [28], HLH in general may benefit from the prompt application of specific therapies to destroy immune cells and suppress inflammation (e.g., with dexamethasone and possibly HLH-specific chemotherapy [24,29]), which may then mitigate unfavorable outcomes like further organ dysfunction and mortality [26,30,31,32].

Case fatality rates for dengue-associated HLH range from 4.5% when active surveillance was conducted [25], to 14.6% in a systematic review of published reports [32], and to 43% in an adult intensive care unit series [33]. To mitigate high mortality, early identification and familiarity with dengue-associated HLH are imperative for prompt treatment by clinicians. This paper therefore aims to review the current clinical and therapeutic knowledge on dengue-associated HLH, and act as a resource to allow clinicians to improve their management of HLH associated with severe dengue.

## 2. Pathophysiology and Presentation of Dengue-Associated HLH

In dengue-associated HLH, dengue infection triggers dysregulated macrophage activation that in turn leads to excessive cytokine secretion (i.e., a “cytokine storm”) and aggressive immune-mediated tissue destruction [34,35,36] (Figure 1).

Fever, bicytopenia (particularly anemia and thrombocytopenia), hepatomegaly, hepatitis, and hyperferritinemia (>500 mcg/L) may occur in both severe dengue and HLH. However, unlike patients with severe dengue alone, patients with HLH may commonly present with prolonged fever of >7 days, splenomegaly, elevated lactate dehydrogenase, hypertriglyceridemia, hypofibrinogenemia, and hemophagocytosis (histiocyte/macrophage engulfing erythrocytes, leukocytes, and platelets, as demonstrated on bone marrow examination, Figure 2) [15,16,20,23,26,37,38,39,40]. The diagnosis of HLH requires five of the following eight criteria as outlined by the HLH-2004 study: (1) fever, (2) splenomegaly, (3) bicytopenia, (4) hypertriglyceridemia (≥3 mmol/L) and/or hypofibrinogenemia (≤1.5 g/L), (5) hemophagocytosis in bone marrow/spleen/lymph nodes, (6) low/absent NK-cell-activity, (7) hyperferritinemia (≥500 mcg/L), and (8) high soluble CD25 (interleukin-2-receptor) level ≥2400 U/mL [41,42,43].

## 3. Identification and Treatment of Dengue-Associated HLH in Adults and Children > 12 Years of Age

Table 1 contains case reports or series of 39 non-pregnant patients with dengue-associated HLH. Persistent or recurrent fever over a duration exceeding 7 days was the main presenting feature for most patients. Few patients had comorbid medical conditions; these included diabetes mellitus [44,45,46], hypertension [47], and obesity [48]. Patients were diagnosed using the criteria outlined by the HLH-2004 study [41], with 33 cases (84.6%) reporting hemophagocytosis on bone marrow aspirate (1 case was determined postmortem [49]), 1 case reported a false negative finding [50]), and no cases mentioned any procedural complications or difficulties.

Treatment of dengue-associated HLH was usually with short courses (about 3–4 days) of high-dose intravenous steroids, e.g., dexamethasone 10 mg/m^2^ daily [51], with some patients tapering steroids over 2–8 weeks [40,50,52,53,54,55]. One case initiated and continued dexamethasone orally for a total of 26 days [56]. Other agents like intravenous immunoglobulin [57,58,59] or etoposide [60,61] were only occasionally used, which departs from the HLH-2004 study treatment protocol. In some reports, good clinical outcomes were achieved by using IV immunoglobulin alone, without steroids [57]. In addition, no other chemotherapy or hematopoietic stem cell transplantation was reported. With steroids alone, no study reported an increased risk of secondary bacterial or fungal infection.

One case involving a pregnant patient with dengue-associated HLH has been reported [62]. The patient was a 28-year-old woman at 38 weeks gestation and had gestational diabetes, controlled by the diet [62]. Dengue-associated HLH was identified based on fever for three days, persistent anemia despite blood transfusion, thrombocytopenia, hepatosplenomegaly from day six with hepatitis, high lactate dehydrogenase (759 IU/L), high ferritin (6930 mcg/L), high triglyceride (4.3 mmol/L), and hemophagocytosis on bone marrow aspirate. The main treatment administered was intravenous methylprednisolone 1 g for three days, followed by oral prednisolone 1 mg/kg for two weeks, and steroid taper over six weeks. Despite severe postpartum hemorrhage requiring massive transfusion, the patient made a complete recovery. Based on this case, pregnancy status does not appear to confer any substantial differences in diagnosis or treatment of dengue-associated HLH, although there may be an increased bleeding risk.

**Table 1 pathogens-13-00332-t001:** Selected reports of dengue-associated HLH in non-pregnant adults and children > 12 years old.

Patient Characteristics (Reference)	Clinical Presentation and Identification of HLH	Treatment and Outcome
25-year-old male (Acharya [60])	Fever for 9 days, hepatosplenomegaly, hepatitis, pancytopenia, high ferritin (1121 mcg/L), high triglyceride (3.1 mmol/L), low fibrinogen (0.9 g/L), hemophagocytosis on bone marrow aspirate	IV dexamethasone 10 mg/m^2^ daily. IV etoposide twice a week for 2 weeks and once weekly for 6 weeks. Clinical recovery on day 8 of hospitalization. Complete recovery at 3 months of follow-up
26-year-old male, no comorbid diseases (Arshad [63])	Fever for 2 weeks, pancytopenia, hepatitis, high LDH, high triglyceride (3.2 mmol/L), high ferritin (24,459 mcg/L), hemophagocytosis on bone marrow aspirate	Supportive care without steroids. Death at hospitalization day 3
23-year-old female, 33-year-old man, both without comorbid conditions (Chang [51])	Persistent and recurrent fever over >5 days, thrombocytopenia, hepatitis, high ferritin (8364–38,068 mcg/L), hemophagocytosis on bone marrow aspirate	IV dexamethasone 10 mg/m^2^ daily for 3–4 days, without any steroid taper. Both patients recovered fully
30-year-old man, microcytic anemia of unknown cause (Cheo [49])	Fever for 3 days, thrombocytopenia, hepatitis, high triglyceride (8.7 mmol/L), high ferritin (>40,000 mcg/L), hemophagocytosis on postmortem bone marrow	Supportive treatment without steroids. Death
33-year-old female, no comorbid diseases (Chung [52])	Fever for 3 days, splenomegaly, neutropenia, thrombocytopenia, hepatitis, high LDH (1775 U/L), high triglyceride (3.5 mmol/L), high ferritin (25,107 mcg/L)	IV dexamethasone 10 mg/m^2^ daily. Dexamethasone tapered over 2 weeks. Full recovery after 18 days of hospitalization
21-year-old female, no comorbid diseases (De Koninck [57])	Fever, leukopenia, thrombocytopenia, hepatitis, high ferritin (8208 mcg/L), hemophagocytosis on bone marrow aspirate	IV immunoglobulin G 1 g/kg/day body weight for 2 days. Full recovery after 14 days
45-year-old female, with hypertension and diabetes mellitus (Ishak [44])	Fever for 9 days, leukopenia, thrombocytopenia, hepatitis, high LDH (3627 U/L), high ferritin (31,013 mcg/L)	IV dexamethasone given, with improvement, hospital discharge on day 19 of illness, and normalized liver function on day 26
17, 19, 32-year-old males, all without comorbid diseases (Jamaludin [40])	Fever for 2–5 days, anemia, thrombocytopenia, lymphopenia, hepatitis, high LDH (1054–5101 U/L), hepatomegaly, splenomegaly, high ferritin (17,432 to >40,000 mcg/L), high triglyceride (1.54–10.4 mmol/L), hemophagocytosis on bone marrow aspirate	For the first case, IV dexamethasone 10 mg/m^2^ daily, IV immunoglobulin 0.5 g/kg. Dexamethasone tapered over 2 weeks. For the second case, dexamethasone 4 mg TDS was administered for 2 days. For the third case, dexamethasone 4 mg TDS was administered for 1 day. All 3 patients achieved complete recovery
18-year-old male, no comorbid diseases (Jha [50])	Fever >3 weeks, hepatosplenomegaly, pancytopenia, high ferritin (10,550 mcg/L), high triglyceride (5.8 mmol/L), but no hemophagocytosis on bone marrow aspirate (i.e., false negative)	IV dexamethasone 10 mg/m^2^ daily (16 mg in this case) for 5 days, followed by oral dexamethasone taper over 3 weeks. Full recovery after 3 weeks
33-year-old male, no comorbid diseases (Lu [64])	Fever, thrombocytopenia, anemia, splenomegaly, hepatitis, high LDH (1243 U/L), hemophagocytosis on bone marrow aspirate (NB. No ferritin levels reported. Only 4 of the 8 HLH-2008 criteria met)	Supportive care without steroids. Complete recovery at 1 month of follow-up
47-year-old male, diabetes mellitus. (Mizutani [45])	Fever for 3 days, hepatitis, leucopenia, thrombocytopenia, high ferritin (9840 mcg/L), low fibrinogen (1.4 g/L), hemophagocytosis on bone marrow aspirate	Supportive care without steroids. Discharged after 9 days of hospitalization with complete recovery of blood counts. Remained well at 1 year of follow-up
63-year-old female with asthma, chronic hepatitis B infection and hemoglobin H disease (Munshi [65])	Fever, hepatomegaly, hepatitis, pancytopenia, high ferritin (>40,000 mcg/L), low fibrinogen (1.9 g/L), hemophagocytosis on bone marrow aspirate	Supportive care without steroids. Improvement and recovery
28-year-old female, no comorbid diseases (Narayanasami [53])	Fever for 6 days, hepatosplenomegaly, anemia, thrombocytopenia, hepatitis, high ferritin (72,100 mcg/L), hemophagocytosis on bone marrow aspirate	IV dexamethasone 10 mg/m^2^ daily for 3 days. Improvement with discharge after 11 days of hospitalization
20-year-old male, 32-year-old male, both without comorbid conditions (Padmaprakash [58])	Fever >7 days, hepatosplenomegaly, pancytopenia, hepatitis, high triglyceride (3–4 mmol/L), high ferritin (18,540 to >24,000 mcg/L), hemophagocytosis on bone marrow aspirate	IV immunoglobulin in both cases, IV dexamethasone in 1 case. Both survived
17-year-old male, no comorbid diseases (Porel [61])	Fever for 2 weeks, pancytopenia, hepatosplenomegaly, hepatitis, high LDH (8690 U/L), high triglyceride (3.3 mmol/L), high ferritin (1680 mcg/L), hemophagocytosis on bone marrow aspirate	Dexamethasone and etoposide following HLH-94 protocol. Significantly improved and discharged well from hospital
17-year-old male, no comorbid conditions (Pradeep [54])	Fever for >4 days, hepatosplenomegaly, hepatitis, persistent pancytopenia, high triglyceride (3.5 mmol/L), high ferritin (>3000 mcg/L), hemophagocytosis on bone marrow aspirate	IV dexamethasone 10 mg/m^2^ daily (17 mg/day in this patient) for 2 weeks, with steroid taper over 8 weeks. The fever settled within 24 h after starting IV dexamethasone. Complete recovery at 8 weeks follow-up
24-year-old female, no comorbid diseases (Ray [55])	Recurrent fever over 1 week, hepatosplenomegaly, anemia, leukopenia, thrombocytopenia, hepatitis, high LDH (2461 U/L), high triglyceride (6.0 mmol/L), high ferritin (2161 mcg/L), hemophagocytosis on bone marrow aspirate	IV dexamethasone 10 mg/m^2^ daily (16 mg/day in this patient), with steroid taper over 8 weeks, followed by dexamethasone pulses to maintain remission of HLH over weeks 9–40. Resolution and no reactivation at 6-month follow-up
33-year-old man, no comorbid conditions (Ren [56])	Fever for 9 days, hepatomegaly, thrombocytopenia, hepatitis, high ferritin (65,212 mcg/L), hemophagocytosis on bone marrow aspirate	Oral dexamethasone 20 mg daily for 2 weeks, followed by 10 mg daily for 4 days, 4 mg daily for 5 days, and then 2 mg daily for 3 days. Improved and discharged after 11 days of hospitalization. In remission at 1 month follow-up
25-year-old female, 38-year-old man and 44-year-old female; all without comorbid diseases (Ribeiro [66])	Fever, hepatosplenomegaly, leukopenia, thrombocytopenia, hepatitis, high LDH (779–1624 U/L), high ferritin (7093–23,451 mcg/L), high triglyceride (3.1–4.6 mmol/L), hemophagocytosis on bone marrow aspirate	Corticosteroids were given to all 3 patients. IV immunoglobulin given to 1 patient. All recovered after 2 weeks of treatment
63-year-old female, Crohn’s disease treated with mercaptopurine and mesalamine, uterine cancer treated with hysterectomy, hypertension, hyperlipidemia, coronary artery disease, obesity, depression, previous thyroidectomy (reason not stated) (Sharp [47])	Fever, leukopenia, thrombocytopenia, anemia, hepatitis, high LDH (727 U/L), high ferritin (>7500 mcg/L), low fibrinogen (<0.6 g/L), hemophagocytosis on bone marrow aspirate	Supportive therapy without steroids. Death at hospitalization day 12 from fulminant multiorgan failure
14-year-old male, no comorbid conditions (Takkinsatian [67])	Fever >7 days, hepatosplenomegaly, leukopenia, thrombocytopenia, high ferritin (>40,000 ng/L), low fibrinogen (1.49 g/L), hemophagocytosis on bone marrow aspirate	IV dexamethasone and IV immunoglobulin. Full recovery after 2 weeks
16- to 43-year-old, 4 females, 2 males, one with previously undiagnosed diabetes mellitus (Tan [46])	Persistent/recurrent fever >7 days, hepatomegaly, hepatitis, pancytopenia/bicytopenia, high ferritin (28,060–66,036 mcg/L), hemophagocytosis on bone marrow aspirate done for 3 patients. One patient also had high triglyceride and low fibrinogen levels	Supportive therapy without steroids in 2 patients. IV methylprednisolone in 4 patients (1 also had IV immunoglobulin G) with prednisolone taper during recovery. Recovery/improvement in 5 of 6 patients at hospital discharge. One patient died (a 43-year-old woman with previously undiagnosed diabetes mellitus)
2 cases, 12 and 14 years old, 1 had beta-thalassemia major and 1 had trisomy 21 (Thadchanamoorthy [68])	Persistent fever for >5 days, anemia, thrombocytopenia, hepatosplenomegaly, hepatitis, high LDH (1280–2106 U/L), high ferritin (6000–32,000 mcg/L), low fibrinogen (1.2 g/L), high triglyceride (2.8–3.2 mmol/L), hemophagocytosis on bone marrow aspirate	Steroid only for 1 patient. IV dexamethasone with IV immunoglobulin for 1 patient. Both recovered
53-year-old male, no comorbid diseases (Wong [69])	Fever, hepatosplenomegaly, hepatitis, anemia, thrombocytopenia, hemophagocytosis on bone marrow aspirate (NB: No ferritin levels reported. Only 4 of the 8 HLH-2008 criteria met)	“Systemic steroid” administered for 2 days prior to presentation. Complete recovery 1 week after presentation
14-year-old male, 35-year-old female with morbid obesity, 56-year-old man (Yew [48])	Fever for >3 days, bicytopenia, hepatitis, high LDH (2719 U/L), high triglyceride (3.2 mmol/L), high ferritin (35,023–93,026 mcg/L), hemophagocytosis on bone marrow aspirate	Supportive therapy without steroids for 1 patient. IV methylprednisolone 500 mg daily for 1–2 days for 2 patients. Complete recovery or improvement for all 3 patients

ICU: intensive care unit. IV: intravenous. LDH: lactate dehydrogenase.

## 4. Identification and Treatment of Dengue-Associated HLH in Young Children up to 12 Years of Age

Table 2 contains case reports or series of 97 young children with dengue-associated HLH. The clinical presentation of young children resembled that of adults. Persistent or recurrent fever over a duration exceeding 7 days was the main presenting feature for most patients. Comorbid conditions were few, and included diabetes, obesity, asthma, and congenital heart disease [25,48]. Like adults, young children were diagnosed using the criteria outlined by the HLH-2004 study [41]. However, unlike adults, about two-thirds of these cases (63 patients) did not report the performance of bone marrow aspiration.

Treatment of dengue-associated HLH in young children appeared to involve either steroids alone [48,70,71], immunoglobulin alone [72], or a combination of steroids and immunoglobulin [67]. Other agents like etoposide [73] were only occasionally used. No other chemotherapy or hematopoietic stem cell transplantation was reported. With steroids alone, or with combination steroid and immunoglobulin therapy, only one study suggested an increased risk of secondary bacterial infection [74].

**Table 2 pathogens-13-00332-t002:** Selected reports of dengue-associated HLH in young children up to 12 years of age.

Patient Characteristics (Reference)	Clinical Presentation and Identification of HLH	Treatment and Outcome
8-month-old male, no comorbid conditions (Arora [73])	Recurrent fever >7 days, hepatosplenomegaly, hepatitis, high LDH (9116 U/L), bicytopenia, high ferritin (>40,000 mcg/L), low fibrinogen (0.76 g/L), high triglyceride (3.1 mmol/L), focal convulsion, embolic infarcts with diffuse leptomeningeal enhancement on MRI	IV dexamethasone 10 mg/m^2^ daily, IV etoposide, intrathecal methotrexate, intrathecal hydrocortisone. Death
7 cases admitted to a pediatric ICU, median age 8 years, 5 (71.4%) males, no comorbid conditions (Bhattacharya [75])	Fever for mean of 5 days, hepatomegaly, thrombocytopenia (mean platelet count 23,000/mm^3^), anemia (mean hemoglobin 8.1 g/dL), hepatitis, hemophagocytosis with bone marrow aspiration in all 7 cases	Steroids were given to 4 patients and all survived. No steroids given to 3 patients, with 2 deaths (28.6% mortality among all 7 cases)
22 cases found from active case surveillance among hospitalized children with dengue in Puerto Rico, median age 1 year, 12 (55%) male, 3 (13.6%) were premature births, 10 (45.5%) had other chronic medical conditions (Ellis [25])	Fever for median of 8 days, high ferritin (median 18,789 mcg/L), 17 (77.3%) had splenomegaly, 19 (86.4%) had hepatomegaly, 16 (72.7%) had anemia, 20 (90.9%) had thrombocytopenia, 17 (77.3%) had leukopenia, hemophagocytosis in 8 of 14 cases with bone marrow aspiration	Overall, 16 (72.7%) received corticosteroids, 13 (59.1%) received IV immunoglobulin, 8 (36.4%) received etoposide; 1 death out of 22 cases (4.5% case fatality rate)
4-year-old female, no comorbid conditions (Gnanasambandam [72])	Fever for 8 days, hepatomegaly, hepatitis, high triglyceride (2.8 mmol/L), thrombocytopenia, high ferritin (7500 mcg/L), low fibrinogen (1.3 g/L)	IV immunoglobulin 2 g/kg over 48 h. Recovery and hospital discharge
1-week-old male, born at 33 weeks gestation, no comorbid conditions (Krishnappa [74])	Encephalopathy, hepatomegaly, hepatitis, anemia, thrombocytopenia, low fibrinogen (1.3 g/L), high ferritin (34,718 mcg/L), low NK cell activity	IV immunoglobulin 0.5 g/kg once. IV dexamethasone 10 mg/m^2^ daily for 2 weeks. Improved from HLH. Had *Staph hominis* sepsis and *Acinetobacter baumanii* ventilator-associated pneumonia. Eventually recovered and discharged after 74 days of hospitalization
27 cases, median 3 years old (range 4 months to 10 years) (Nandhakumar [13])	Fever for 8 days on average, hepatitis, high ferritin (mean 26,620 mcg/L)	Overall, 18 improved with supportive care alone; 9 cases with hemodynamic instability, respiratory distress, and neurological involvement given steroids and IV immunoglobulin, with 4 deaths
11 cases, 2–10 years of age, 6 male (54.5%) (Parajuli [76])	Fever (100%), hepatomegaly (90.9%), splenomegaly (36.4%), cytopenias (90.9%), high triglyceride (18.2%), low fibrinogen (27.3%), high ferritin (mean 8311 mcg/L), hemophagocytosis (63.6%)	Six received IV dexamethasone, two received IV immunoglobulin. Five deaths (45.5%)
10-year-old male, no comorbid diseases (Ray [77])	Fever for 2 weeks, bicytopenia, hepatosplenomegaly, hepatitis, high LDH (1180 U/L), high triglyceride (3.9 mmol/L), high ferritin (2800 mcg/L)	Steroids and supportive care. Full recovery
15 children, mean age 2 years, no comorbid diseases (Restrepo [26])	Fever for >7 days (mean 10 days), hepatomegaly, anemia, hepatitis, high LDH (mean 4209 U/L), high triglyceride (mean 3.8 mmol/L)	Steroids given to 8 children (53%) with 50% survival, combined steroids and immunoglobulin given to 4 children (27%) with 25% survival (further details about disease severity not stated). Among all 15 cases, 4 deaths (27%)
9 cases, mean age 10.4 years, no comorbid diseases (Singh [71])	Splenomegaly, thrombocytopenia, hepatitis, high LDH (mean 7929 U/L), high ferritin (mean 34,593 mcg/L)	All received steroids; 2 deaths (22.2% mortality)
3-year-old female, 10-year-old boy, both without comorbid conditions (Takkinsatian [67])	Fever for >7 days, leukopenia, thrombocytopenia, hepatitis, high ferritin (>40,000 mcg/L), hemophagocytosis with bone marrow aspiration in both cases	IV dexamethasone 10 mg/m^2^ and IV immunoglobulin 0.5 g/kg daily. Full recovery after 1–2 weeks
9-year-old female with obesity (Yew [48])	Fever for >7 days, thrombocytopenia, hepatitis, high ferritin (97,316 mcg/L), low fibrinogen (0.95 g/L)	IV methylprednisolone 500 mg BD for 1 day. Full recovery on day 11 of illness

ICU: intensive care unit. IV: intravenous. LDH: lactate dehydrogenase. MRI: magnetic resonance imaging.

## 5. Challenges in the Diagnosis and Treatment of Dengue-Associated HLH

Dengue-associated HLH is often associated with severe dengue, which is dengue infection with any of the following: severe plasma leakage (leading to shock, fluid accumulation with respiratory distress), severe bleeding, severe organ involvement (aspartate aminotransferase or alanine aminotransferase ≥1000 U/L, impaired consciousness, or other organ failure) [10]. The clinical features of severe dengue can mask the identification of HLH, which means that clinicians need to maintain a high index of suspicion and actively look for the presence of any of the eight HLH-2004 criteria. Fortunately, assessment of most of these criteria should be available in routine clinical practice. Some other clinical features can help distinguish severe dengue from dengue-associated HLH, as these are more commonly found in the latter: (1) prolonged or recurrent fever of >7 days [32,50] (consider also co-infection, which occurred in 10.4% of 122 cases [32]), and (2) anemia without intravascular hemolysis or massive bleeding [32,75].

Bone marrow examination is not necessary when enough criteria have been fulfilled [32], although bone marrow aspiration seems to be safe in many reports involving adults and older children with dengue-associated HLH, consistent with the low 0.08% adverse event rate found in an audit performed by the British Society for Haematology across 63 hospitals [70]. Additionally, when it is available, specialized testing for elevated interleukin-2 receptor (i.e., CD25) can help distinguish dengue-associated HLH from severe dengue alone [71].

For the treatment of dengue-associated HLH, no randomized trials are available, which makes treatment selection both challenging and uncertain. Nonetheless, published experience from contemporary cases mainly relies on supportive care with or without the use of short courses (3–4 days) of high-dose steroids [51]. Although some patients had a favorable outcome without steroids [45,64], the high risk of mortality and the relatively low risk of short courses of steroids argue for using steroids to treat dengue-associated HLH. Another reason to support the use of short courses of high-dose steroids is that there is no apparent way of predicting the self-limiting trajectory of HLH with supportive management alone.

Specific steroid treatment could be IV methylprednisolone or dexamethasone alone, without the need for IV immunoglobulin or chemotherapy. Dexamethasone may be preferred when HLH involves the central nervous system, as it has superior penetration across the blood–brain barrier compared to prednisolone [78]. Dexamethasone may even be initiated orally [56], given its good bioavailability of 81% [79]. Similarly, prednisone and prednisolone have good oral bioavailability of >80% [59], although their upfront use in dengue-associated HLH has not been reported.

Some authors have suggested combined therapy of steroids with immunoglobulin [59], an example being methylprednisolone 1 g/day for 3–5 days and immunoglobulin 1 g/kg/day for 2 days [80]. In addition, clinicians can consider IV immunoglobulin [58] and etoposide as rescue therapies, and use IV immunoglobulin as an alternative in case steroids cannot be given [57,81]. Dengue-associated HLH should not require chemotherapy or hematopoietic stem cell transplantation. Finally, for persistently or recurrently febrile patients, it remains important to consider and promptly treat bacterial co-infection, which occurred in 33% of dengue-associated HLH cases [26].

The clinical management of dengue-associated HLH is summarized in Table 3. When treatment is effective, gradual recovery with complete resolution of clinical and laboratory abnormalities within 2–8 weeks is expected [44,48,50,52,54,56,57,62,64,66,67,69].

## 6. Future Directions

Future research is required to uncover the prognostic factors in dengue-associated HLH per se, and to identify individuals who may continue to benefit from supportive treatment alone (i.e., without HLH-specific treatment like steroids). In addition, while there are numerous case reports and series, the epidemiology of dengue-associated HLH remains unclear. Given the relatively rarity of dengue-associated HLH, especially for important special populations like pregnant women, studying its epidemiology would likely require a global registry.

Additionally, the treatment of dengue-associated HLH appears to be less intensive than for other forms of HLH (e.g., familial HLH), but optimal specific treatment remains unknown. While conventional randomized clinical trials of multiple treatment options would be difficult to achieve for a rare condition, an adaptive trial design may be used to efficiently study the effects of treatment with fewer patients and shorter clinical investigation times [82]. In the case of dengue-associated HLH, such an adaptive trial design may be applied in the comparison of supportive treatment alone, steroids, intravenous immunoglobulin, and chemotherapy, with regards to improving organ dysfunction and mortality.

Novel agents that have been used in non-dengue-associated HLH could be further studied in the context of dengue-associated HLH. Possible medications include emapalumab, a human anti-interferon-gamma antibody [83], tocilizumab, a monoclonal antibody targeting the interleukin-6 receptor [84], and anakinra, a recombinant interleukin-1 receptor antagonist [85], although these may increase the risk of secondary infection. Another promising agent is ruxolitinib [86], an orally administered Janus kinase inhibitor that has been shown to selectively dampen the cytokine storm in primary HLH while preserving the immune response against infection.

## 7. Conclusions

Dengue, a significant global health concern, lacks specific treatments beyond supportive care, leaving room for improved outcomes. While efforts have primarily focused on managing capillary permeability and bleeding, the presence of dengue-associated HLH demands attention. Characterized by dysregulated macrophage activation and cytokine storm akin to sepsis, dengue-associated HLH presents challenges in diagnosis and management. Despite its rarity, dengue-associated HLH significantly elevates mortality rates in severe dengue cases. Diagnosis relies on fulfilling HLH-2004 criteria, while treatment typically involves short courses of high-dose intravenous steroids. Notably, outcomes can be favorable with steroid therapy alone. Future research should focus on prognostic factors and treatment optimization, potentially through adaptive trial designs. Enhanced understanding and management of dengue-associated HLH could significantly improve clinical outcomes and inform global dengue management strategies.

## Figures and Tables

**Figure 1 pathogens-13-00332-f001:**
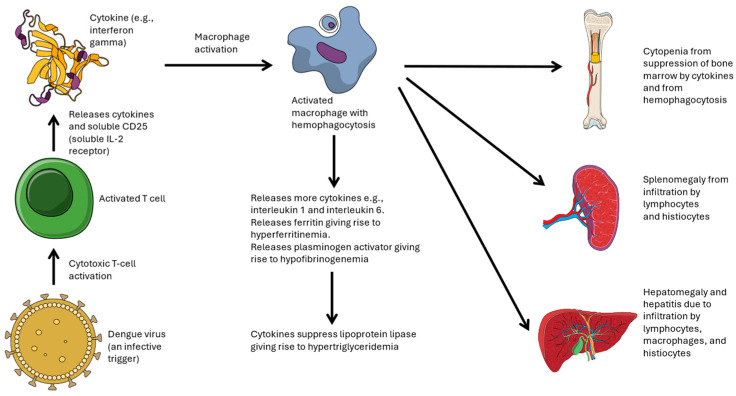
Pathophysiology of dengue-associated hemophagocytic lymphohistiocytosis. Dengue virus infection triggers the activation of cytotoxic T-cells, leading to cytokine release, macrophage activation, and subsequent dysfunction of vital organs. Images from Science Figures (open license) (accessed from https://sciencefigures.org on 30 March 2024). Figure constructed based on information from Hines et al. [35] and Keenan et al. [36].

**Figure 2 pathogens-13-00332-f002:**
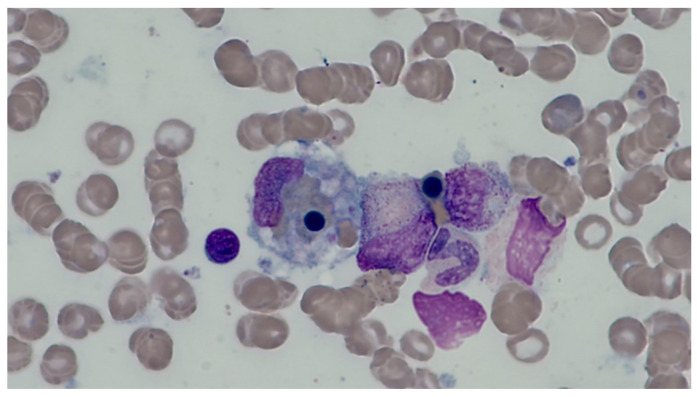
Bone marrow smear with hemophagocytosis. The large cell in the center of the image is a macrophage that has phagocytosed several red blood cells and platelets. Image by Petter Quist-Paulsen (license CC BY SA 3.0) (accessed from https://sml.snl.no/hemofagocytisk_lymfohistiocytose_-_HLH on 30 March 2024).

**Table 3 pathogens-13-00332-t003:** Clinical management of dengue-associated HLH.

Management Step	Initial Actions	Further Actions
Diagnosis	Consider HLH routinely in all cases of severe dengue. In addition, suspect HLH based on the presence of any of the following in a patient with dengue: (1) prolonged or recurrent fever of >7 days; (2) anemia without intravascular hemolysis or massive bleeding	Send blood tests for triglycerides, fibrinogen, and ferritin. Send blood tests for NK-cell-activity and high-soluble interleukin-2-receptor if available. Perform bone marrow aspirate for hemophagocytosis if possible. The presence of at least 5 of the 8 HLH-2004 criteria confirms HLH *. Rule out other associations of HLH (other infections, autoimmune diseases, and malignancy)
Treatment	Maintain adequate hydration. Correct anemia. Correct thrombocytopenia if bleeding. Supportive care for organ dysfunction. If bacterial co-infection is suspected, then obtain blood cultures and administer broad-spectrum antibiotics promptly	Consider IV methylprednisolone 1 g/kg or IV dexamethasone 10 mg/m^2^ daily for 3–4 days or more, contingent on improvement of clinical and laboratory parameters. Steroids may be oral and tapered over 2 or more weeks, contingent on improvement of clinical and laboratory parameters. Consider IV immunoglobulin and/or etoposide as rescue therapy

* Diagnosis of the HLH requires 5 of the following 8 criteria as outlined by the HLH-2004 study: (1) fever, (2) splenomegaly, (3) bicytopenia, (4) hypertriglyceridemia (≥3 mmol/L) and/or hypofibrinogenemia (≤1.5 g/L), (5) hemophagocytosis in bone marrow/spleen/lymph nodes, (6) low/absent NK-cell-activity, (7) hyperferritinemia (≥500 mcg/L) and (8) high soluble CD25 (interleukin-2-receptor) level ≥2400 U/mL. IV: intravenous.

## Data Availability

All data are available in the published manuscript.

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
