# Peer review of "Dengue-Associated Hemophagocytic Lymphohistiocytosis: A Narrative Review of Its Identification and Treatment"

_pathogens, 2024, doi:10.3390/pathogens13040332_

Round 1

Reviewer 1 Report

Comments and Suggestions for Authors

The paper offers a comprehensive review of the latest clinical and therapeutic insights concerning dengue-associated HLH. The manuscript is well-crafted, with current and relevant information. However, it shares the inherent limitations of any narrative review, which, in both my opinion and that of many others, tends to exhibit a significant degree of bias and subjectivity [1].

Methods (Page 3, Lines 86-91):

·      Although narrative reviews do not typically require a detailed description of the search strategy, the author has provided the time frame and keywords used for the search. Despite this, the article exhibits substantial methodological shortcomings, common to narrative reviews, such as a high risk of bias and subjectivity. A more detailed search algorithm, including MeSH terms, should be disclosed for enhanced transparency.

·      The review is limited by its reliance on a single database, a frequent criticism of narrative reviews.

·      It would be beneficial to expand on the selection criteria for the included studies and detail any quality assessments conducted to ascertain the robustness of the review's conclusions.

References

1.     Wilczynski, S.M. (2017). Other Sources of Evidence. A Practical Guide to Finding Treatments That Work for People with Autism, pp.13–19. doi: https://doi.org/10.1016/b978-0-12-809480-8.00002-9.

Author Response

Reply to the Review Report (Reviewer 1)

The paper offers a comprehensive review of the latest clinical and therapeutic insights concerning dengue-associated HLH. The manuscript is well-crafted, with current and relevant information. However, it shares the inherent limitations of any narrative review, which, in both my opinion and that of many others, tends to exhibit a significant degree of bias and subjectivity [1].

References

  1. Wilczynski, S.M. (2017). Other Sources of Evidence. A Practical Guide to Finding Treatments That Work for People with Autism, pp.13–19. doi: https://doi.org/10.1016/b978-0-12-809480-8.00002-9.

Reply: Thank you for your feedback. I acknowledge these limitations and have addressed these in a new section “Limitations of this review”.

Methods (Page 3, Lines 86-91):

  • Although narrative reviews do not typically require a detailed description of the search strategy, the author has provided the time frame and keywords used for the search. Despite this, the article exhibits substantial methodological shortcomings, common to narrative reviews, such as a high risk of bias and subjectivity. A more detailed search algorithm, including MeSH terms, should be disclosed for enhanced transparency.

Reply: The search was done to support the narrative review with more contemporary articles, and this description has been added to the last paragraph of the Introduction. No MeSH terms were used. The methodological shortcomings are included as limitations in a new section “Limitations of this review”.

  • The review is limited by its reliance on a single database, a frequent criticism of narrative reviews.

Reply: This has been included as a limitation in a new section “Limitations of this review”.

  • It would be beneficial to expand on the selection criteria for the included studies and detail any quality assessments conducted to ascertain the robustness of the review's conclusions.

Reply: No quality assessments were conducted as all relevant studies were included in the review. The lack of quality assessment is included as a limitation in a new section “Limitations of this review”.

Reviewer 2 Report

Comments and Suggestions for Authors

In this narrative review, Dr. See discussed dengue-associated hemophagocytic lymphohistiocytosis (HLH). Overall, this is a nicely written manuscript. I enjoyed reading this piece of work. Nonetheless, I do have some comments:

1) Because this is a narrative review, methods section is not needed. If this is a scoping or systematic review, the author must adhere with the PRISMA guideline and register the protocol in PROSPERO. Please clarify the design of this study. 

2) Regarding this statement "Overall management of dengue is supportive", I think it is important to also briefly acknowlegde the presence and perhaps discuss dengue vaccines. 

3) Since HLH could potentially lead to cytokine storm, what about the potential applications of immunomodulatory agents as used in the setting of COVID-19 (PMID: 34224330). 

4) Please clarify whether the case reports in Table 1 are "selected" or "scoped" from the databases. If they are selected, please specify the selection criteria and the reasons for using those criteria. 

5) Since Table 2 only contains a single report, I think it would not be useful to present it as a separate Table. Consider merging this with table 1 or just describe in text. 

6) Based on the comparison between non-pregnant vs. pregnant patients, what is the take-home messages? Are they truly indifferent? It seems that the bleeding risk is increased based on the statement in Table 2 "Severe postpartum hemorrhage requiring massive transfusion."?

7) I think it would be informative to add the type of study of those reports listed in Table 1-3. Are they all case reports?

8) I think Section 5 is redundant because the management has been explained in the previous sections? Please clarify and adjust to avoid redundance.

9) Please add discussion about the existing challenge in HLH diagnosis and treatment before Section 6 about Future directions.

10) I think it is insightful to add an illustrative diagram depicting the pathophysiology of dengue-associated HLH. Please add.

Comments on the Quality of English Language

No comment

Author Response

Reply to the Review Report (Reviewer 2)

In this narrative review, Dr. See discussed dengue-associated hemophagocytic lymphohistiocytosis (HLH). Overall, this is a nicely written manuscript. I enjoyed reading this piece of work. Nonetheless, I do have some comments:

Reply: Thanks for your feedback. My replies are elaborated below.

1) Because this is a narrative review, methods section is not needed. If this is a scoping or systematic review, the author must adhere with the PRISMA guideline and register the protocol in PROSPERO. Please clarify the design of this study.

Reply: I have removed the methods section and clarified that this study is a narrative review.

2) Regarding this statement "Overall management of dengue is supportive", I think it is important to also briefly acknowlegde the presence and perhaps discuss dengue vaccines.

Reply: Added a paragraph in the Introduction about dengue vaccines.

3) Since HLH could potentially lead to cytokine storm, what about the potential applications of immunomodulatory agents as used in the setting of COVID-19 (PMID: 34224330).

Reply: Added a paragraph in the section “Future directions” about novel immunomodulatory agents.

4) Please clarify whether the case reports in Table 1 are "selected" or "scoped" from the databases. If they are selected, please specify the selection criteria and the reasons for using those criteria.

Reply: The case reports were included from all publications published over the last 5 years to support the review with more contemporary articles and to reflect current management strategies.

5) Since Table 2 only contains a single report, I think it would not be useful to present it as a separate Table. Consider merging this with table 1 or just describe in text.

Reply: Information for Table 2 is now described in the text.

6) Based on the comparison between non-pregnant vs. pregnant patients, what is the take-home messages? Are they truly indifferent? It seems that the bleeding risk is increased based on the statement in Table 2 "Severe postpartum hemorrhage requiring massive transfusion."?

Reply: The treatment appeared to be similar, though I agree that bleeding risk seemed to be elevated.

7) I think it would be informative to add the type of study of those reports listed in Table 1-3. Are they all case reports?

Reply: Yes, these are all case reports or series.

8) I think Section 5 is redundant because the management has been explained in the previous sections? Please clarify and adjust to avoid redundance.

Reply: Redundant information has been removed. The parts retained contain new information that is now used in a new section about existing challenges in HLH diagnosis and treatment. The new information comprises the following points:

  1. Discussion about the need for bone marrow examination
  2. Discussion about the need and type of steroids
  3. Discussion about the role of combined therapy
  4. A summary table, which may be useful as a quick reference for clinical practice.

9) Please add discussion about the existing challenge in HLH diagnosis and treatment before Section 6 about Future directions.

Reply: Added a new section about existing challenges in HLH diagnosis and treatment.

10) I think it is insightful to add an illustrative diagram depicting the pathophysiology of dengue-associated HLH. Please add.

Reply: Added an illustrative diagram to depict the pathophysiology of dengue-associated HLH.

Round 2

Reviewer 2 Report

Comments and Suggestions for Authors

Thanks for the response, I do have a few pertinent issues:

"A search of PubMed® (pubmed.ncbi.nlm.nih.gov, accessed and updated on 30 March 102 2024) was performed using the term “dengue AND (hemophagocytic OR haemophagocytic) AND lymphohistiocytosis”, which yielded 43 articles published over the last 5 years. This was done to update the author’s personal library with more contemporary articles. All articles relevant to the considerations covered in this narrative review were included."

"To support this narrative review with more contemporary articles and to reflect current management strategies, a search of PubMed® (pubmed.ncbi.nlm.nih.gov, accessed and updated on 30 March 2024) was performed using the term “dengue AND (hemophagocytic OR haemophagocytic) AND lymphohistiocytosis” was done to uncover recent cases and series published over the last 5 years."

Please remove such statements as they represent that the author performed the study systematically. Please check the manuscript carefully as their presence is inappropriate in a narrative review. 

"Figure 1. Images from Science Figures (open license) (accessed from https://sciencefigures.org on 14 April 2024). Figure constructed based on information from Hines et al [35] and Keenan et al [36].

Please add the titles and a short description of this figure.

Section 7 is not necessary. I think readers know about these inherent limitations of all narrative review and they don't need to be declared specifically. If the author claimed that all cases within 5 years were included, this indicates that the author indeed performed a systematic review, instead of a narrative review, which serve for different purposes. 

Again, please carefully decide the type of review and adjust the text accordingly and I have to remind that systematic review is not usually done by a single author as some authors play a role in dispute management and other tasks as well. 

Comments on the Quality of English Language

No comment

Author Response

Reply to the Review Report (Reviewer 2)

Thanks for the response, I do have a few pertinent issues:

"A search of PubMed® (pubmed.ncbi.nlm.nih.gov, accessed and updated on 30 March 102 2024) was performed using the term “dengue AND (hemophagocytic OR haemophagocytic) AND lymphohistiocytosis”, which yielded 43 articles published over the last 5 years. This was done to update the author’s personal library with more contemporary articles. All articles relevant to the considerations covered in this narrative review were included."

"To support this narrative review with more contemporary articles and to reflect current management strategies, a search of PubMed® (pubmed.ncbi.nlm.nih.gov, accessed and updated on 30 March 2024) was performed using the term “dengue AND (hemophagocytic OR haemophagocytic) AND lymphohistiocytosis” was done to uncover recent cases and series published over the last 5 years."

Please remove such statements as they represent that the author performed the study systematically. Please check the manuscript carefully as their presence is inappropriate in a narrative review.

Reply: Thanks for your feedback. Such statements have been removed.

"Figure 1. Images from Science Figures (open license) (accessed from https://sciencefigures.org on 14 April 2024). Figure constructed based on information from Hines et al [35] and Keenan et al [36]."

Please add the titles and a short description of this figure.

Reply: A title and short description of Figure 1 have been included.

Section 7 is not necessary. I think readers know about these inherent limitations of all narrative review and they don't need to be declared specifically. If the author claimed that all cases within 5 years were included, this indicates that the author indeed performed a systematic review, instead of a narrative review, which serve for different purposes.

Again, please carefully decide the type of review and adjust the text accordingly and I have to remind that systematic review is not usually done by a single author as some authors play a role in dispute management and other tasks as well.

Reply: Thanks for your feedback. Section 7 has been removed.